# The Intention and Influence Factors of Nurses' Participation in Telenursing

**Mei-Ying Chang [1],[†], Fang-Li Kuo [1],[†], Ting-Ru Lin [2] , Chin-Ching Li [3],*and Tso-Ying Lee [2],[4],***

1 National Taipei University of Nursing and Health Sciences, Taipei 112, Taiwan;
Lotus3235@yahoo.com.tw (M.-Y.C.); mindurmbhealth@gmail.com (F.-L.K.)
2 School of Nursing, National Taipei University of Nursing and Health Sciences, Taipei 112, Taiwan;
061117003@ntunhs.edu.tw
3 Department of Nursing, Mackay Medical College, New Taipei City 25245, Taiwan
4 Nursing Department of Cheng Hsin General Hospital, Taipei 112, Taiwan
* Correspondence: chinching@mmc.edu.tw (C.-C.L.); tsoyinglee@gmail.com (T.-Y.L.)
† These authors contributed equally to this work.

**Abstract:** This study aimed to identify factors that significantly affect the behavioral intention of nursing staff to practice telenursing, applying the decomposed theory of planned behavior (DTPB) as the research framework. This cross-sectional survey study collected data from a valid sample of 203 responses from nurses from a regional hospital in Taipei City, Taiwan. The results of data analysis showed that nursing staff's attitude, subjective norms, and perceived behavioral control toward telenursing correlated positively with behavioral intention to participate in telenursing. Decomposing the main concepts identified two significant predictive determinants that influence nurses' behavioral intentions: (a) facilitating conditions ($\beta = 0.394$, $t = 5.817$, $p = 0.000 < 0.001$) and (b) supervisor influence ($\beta = 0.232$, $t = 3.431$, $p = 0.001 < 0.01$), which together explain 28.6% of the variance for behavioral intention. The results of this study indicated that support and encouragement from nursing supervisors are important factors affecting nurses' intention to practice telenursing. Education and training, health policies advocacy and the provision of adequate facilitating technologies and recourses are important factors for improving intention to practice telenursing.

**Keywords:** attitude; conscious behavior; nursing; subjective norms; telehealth telenursing

## 1. Introduction

The term telehealth is defined by the Health Resources Services Administration of the United States [1] as "the use of electronic information and telecommunications technologies to support and promote long-distance clinical health care, patient and professional health-related education, public health, and health administration" [1]. Telenursing or telehealth nursing is described as a subset of telehealth with a focus on nursing practice [2]. The American Nurses Association (ANA) has defined telenursing as the use of "technology to deliver nursing care and conduct nursing practices" [3]. The American Telehealth Association refers to telehealth nursing as "a tool for delivering nursing care remotely to improve efficiency and patient access to healthcare" [4].

In Taiwan, nurses have been participating in the telehealth continuum since 1996, mainly using telephone interviews to initiate and coordinate follow-up after discharge, provide guidance and education on self-care at home, as well as to respond to any health inquiries during home-based care after discharge to home [5]. Since then, tele-technologies such as remote patient monitoring, and in some instances mobile devices, have been used to monitor and record patient physiological data and provide general nursing educational information to patients and caregivers. The versatility of functions and roles of telehealth nurses has expanded to include abilities to guide patients to clinical visits, clarify treatment options, educate patients about self-care at home, and assist with appointment scheduling.

A survey of health care resource utilization using the Taiwan National Health Insurance Research Database (NHIRD) (2009 to 2011) found that telehealth care has significantly reduced health care resource utilization [6], particularly for preventable hospitalization and the length of hospital stay. Expanding telehealth services could also improve the quality and advantages of home-based services [7]. During the COVID-19 pandemic, telehealth has emerged as a crucial approach to providing health care access for patients, given its ability to reduce cross-infection risk, reduce costs and improve the quality of care [8]. Nevertheless, successful implementation of telehealth relies on effective operation at the hospital management level, as well as on the acceptance of tele-technologies by medical personnel. The role of nurses is an indispensable part of telehealth implementation and a recent systematic review found that nurses' skills and attitudes could be preventive factors in the implementation of telehealth and telenursing [9]. However, while demand for telehealth care and involvement of nursing staff is increasing, knowledge of factors that influence nurses' intention and willingness to practice telenursing is limited. We hypothesized that the application of the decomposed theory of planned behavior (DTPB) model would assist in identifying influencing factors for nursing staff's acceptance of telenursing, further supporting the expansion of telehealth through increased adoption. The objective of the present study is to identify factors that significantly affect nurses' behavioral intention to practice telenursing, applying the DTPB model as the research framework.

## 2. Literature Review

### 2.1. Factors Influencing the Use of Telenursing by Nursing Staff

Care providers' pre-existing experience and self-confidence in dealing with similar or new features from other technologies may not only affect their attitude about using telehealth technology [10], but also the way in which they use telehealth technology for virtual communication [11]. A recent study showed that exploitative IT use is an important driver to increase virtual service performance, and personal characteristics such as habits are positively associated with both exploitative and exploratory use behaviors, while computer self-efficacy is positively associated with exploitative use of telehealth technology [12].

### 2.2. The Role of Nursing Staff in Telenursing

The role of nursing in telecare was built on the idea of telephone triage. Using the phone, nursing staff will help to determine the most appropriate health care for a particular case after assessing the urgency of the caller's situation. Cases that require immediate intervention or non-urgent cases may still require medical care. The role of nursing is an indispensable part of telemedicine implementation [13]. More than just a phone call, telenursing involves responsibilities ranging from calming the caller to assisting with ambulance dispatch in critical situations [14]. Telenursing is specifically described as telemedicine technology designed to deliver and implement nursing care [15], including providing telecare guidance and supportive care delivery in the form of medication monitoring, tracking, data collection, and pain management [16]. A survey of working locations of telecare-nursing staff in the United States revealed that the most common workplaces were hospitals (25.3%), universities (22.1%), and 18 other locations such as military units, private studios, insurance companies and prisons [17]. The work includes administrative management, consultation, direct care of visiting patients/inpatients, immediate care of remote patients, research, supervision, and teaching [17].

Swedish scholars identified five categories of telenursing services, including: 1. Evaluation, referral, and nursing advice for self-care; 2. Providing support by maintaining contact with the caller until the problem is resolved; 3. Strengthening self-confidence by confirming and praising the caller's skills in self-care behaviors and encouraging increased self-care confidence; 4. Teaching correct nursing skills and health education by phone; and 5. Facilitating the caller's learning process based on the caller's current understanding of symptoms and treatment [18].

Telehealth nurses in Taiwan usually participate in connection, integration, and tracking rather than physical assessment and nursing guidance [16]. Data of telephone interviews and consultation with the case manager of the Discharged Patient Telemedicine Care Center revealed that telehealth deals most often with care problems (62%), vital signs issues (23.8%), and obstacle elimination (12.1%). Work process problems represented 2% of calls and the remainder included post-discharge wound care, tube feeding or other intubation care, and pain management [19].

### 2.3. The Performance of Nursing Staff in Telenursing Services

Depression is a major complication for older adults with chronic diseases who are at home suffering from congestive heart failure and chronic obstructive pulmonary disease. A telehealth nurse records daily symptoms and weight, monitors medications, provides eight lessons per week to treat depression, and opens the communication channel between the patients and their attending physicians. Results after three and six months showed a 50% decrease in depression scores than in those who only received home care and psychological education. The number of emergency visits was also significantly less than that of the other group. In those who dealt with disease-related problems, significant improvements in skills and self-efficacy demonstrate the value of nursing staff in the telecare role [20].

A study in Taipei City, Taiwan, provided free telecare services from April 2009 to December 2014, offering 177,409 calls for citizens' health-related telephone care, and 94.5% of citizens were satisfied with the telecare service [21]. Among the 5747 electrocardiogram telecare services, 5174 regular and active telephone care calls (accounting for 90.0%) were provided. Telecare service and consultation are critical in enhancing patients' self-care ability [22].

Notably, patients who participated in remote patient monitoring (RPM) were less likely to experience hospital stays, incurred fewer ED and urgent-care visits, and reported better management of their symptoms. Increased physical stamina as well as greater overall patient satisfaction and emotional well-being were also noted [23].

A better patient experience with COVID-19 patients has been reported for telehealth services used when family members are not allowed in patient care settings. Examples of telenursing technologies include implementing devices with video capability to provide human interaction, and conversational agent intelligence solutions such as voice-controlled virtual care assistants to help improve communication between the patient, nursing staff and family members [24].

### 2.4. Decomposed Theory of Planned Behavior

Taylor and Todd (1995) [25] developed the "Decomposed Theory of Planned Behavior" (DTPB) based on the technology acceptance model (TAM) [26], the planned behavior theory of Ajzen [27], and other references, defining attitude, subjective norms, and conscious behavior control, as associated with behavioral intention and actual behavior. The decomposed theory well explains the influencing factors of behavioral intention. Previous studies using the DTPB model have demonstrated, for example, nurses' readiness to accept mobile electronic medical record systems [28], and nurses' attitudes towards occupational transformation processes associated with digital care technologies [29]. Nurses' experience with the barriers and facilitators of using telehealth applications has also been explored, and it was found that nurses' face-to-face skills and attitudes were limiting factors in the acceptance of telehealth [9], as well as nurses' perspectives on the efficacy of telenursing and quality of care, which was largely positive, but knowledge was lacking [30].

Figure 1 shows the deconstruction of the theoretical model of projecting behavior, described as follows:

Behavioral Intention: Ajzen proposed the theory of planned behavior, showing that the tendency and level of action indicating that an individual desires to engage in a particular behavior refers to the psychological intensity of the action expressed in the decision process of behavior choice and the psychological intensity of the activity [25,27].

Attitude: According to the decomposed theory of planned behavior, proposed by Taylor and Todd (1995) [25], three factors that promote behavioral intention are: 1. Perceived usefulness, which refers to the level of usefulness users think is possible for a particular system [25,31]; 2. Ease of Use, which refers to the ease of use of a system; and 3. Compatibility, which refers to the level to which innovation meets the current value, experience, and current needs of potential adopters [25,32]. Many studies have revealed that compatibility is one of the keys to the success of telecare [33].

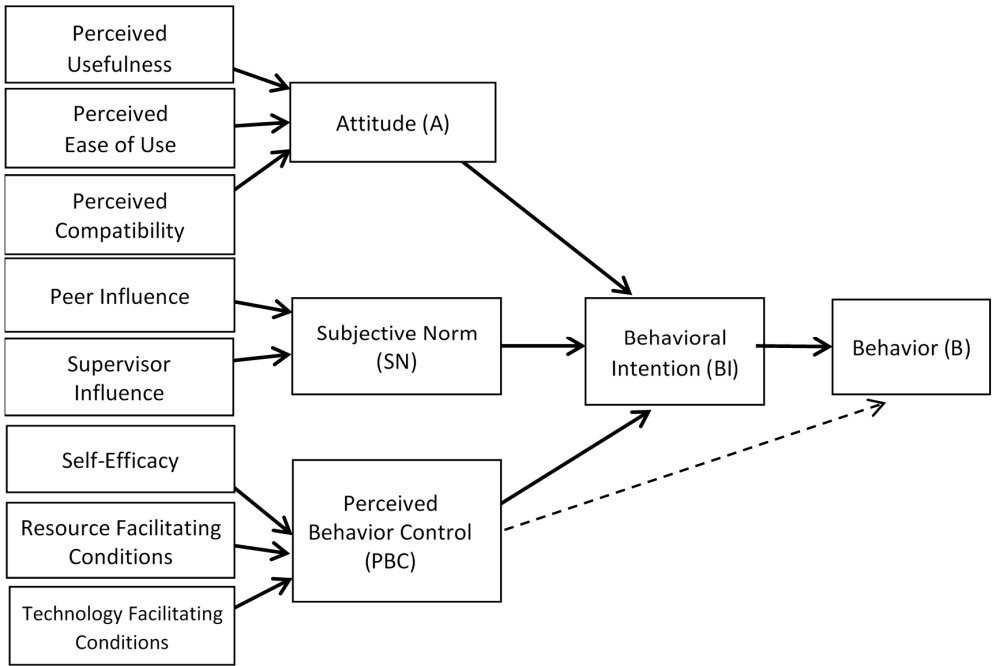

**Figure 1.** Deconstruction of the theoretical model of projecting behavior. Adaptation from: Taylor, S., and Todd, P.A. (1995) Understanding information technology usage: A test competing models. *Information System Research, 6* (2), 144–176 [25].

Subjective Norm: This refers to the social pressure a person feels when projecting a particular behavior to important people such as parents, spouses, friends, and colleagues, etc., as though the behavior is expected. When the positive subjective norm is more robust, it is easier to induce the intention to engage in the behavior. Taylor and Todd (1995) [25] decomposed two parts to subjective norms into the influence of supervisors and peers.

Perceived Behavioral Control: Perceived Behavioral Control illustrates that although individuals control most behaviors by themselves, they do not entirely control other types of behaviors [27]. When the individual owns more power, resources, and opportunities, control ability is sufficient, and the individual adopts the behavior consciously, making control more vital. Taylor and Todd (1995) [25] decomposed perceived behavior control into three parts: self-efficacy, resource facilitating conditions, and technology facilitating conditions. The present study combines resource and technology facilitating conditions as facilitating conditions.

## 2.5. Research Purpose and Framework

This study aimed to identify factors that significantly affect nurses' behavioral intention to practice telenursing, applying the DTPB as the research framework.

As shown in Figure 1, the three primary concepts (i.e., attitude subjective norm, and perceived behavioral control) directly influence behavioral intentions, and can be decomposed into seven dimensional constructs (i.e., perceived usefulness, perceived ease of use, compatibility, peer influence, supervisory influence, self-efficacy, facilitating conditions). Accordingly, the following hypotheses were proposed:

**H1:** *Perceived usefulness, perceived ease of use and/or compatibility have a positive influence on nurses' attitudes toward practicing telenursing.*

**H2:** *Peer influence and/or supervisory influence have a positive influence on nurses' subjective norms towards practicing telenursing.*

**H3:** *Self-efficacy and/or facilitating conditions have a positive influence on nurses' perceived behavioral control toward practicing telenursing.*

Nursing staff's attitude, subjective norms, and perceived behavioral control have a positive influence on their behavioral intention to practice telenursing.

## 3. Research Method

### 3.1. Research Tools

In the present study, the construct measures shown in Figure 1 were adopted from the established decomposed TPB model and factors were measured using a 5-point Likert scale (strongly agree, agree, neutral, disagree, strongly disagree). The questionnaire was self-developed, including structured items based on the established DTPB model (attitude, subjective norms, perceived behavior control, behavioral intentions), and participants' demographics. The questionnaires take approximately 30 min to complete (all items in the questionnaires are available in Supplementary Material Table S1). The target participants were nursing staff in a regional hospital in Taipei City metropolitan area who were included by convenience sampling. A total of 220 questionnaires were distributed by the hospital administrator (considering a response rate of 50% and 10:1 ratio of observed value: independent variable). To ensure the validity and reliability of the questionnaire, all items were evaluated by a panel of nursing experts, and construct validity and reliability were performed.

The study adopted the content validity index of expert validity, and the total content validity index (Total CVI) of the questionnaire was 0.93. The Cronbach's α value was used to evaluate the self-developed tool after collecting the data. The Cronbach's α value of attitude, subjective norms, perceived behavior control, and behavioral intention was 0.948, 0.862, 0.790, and 0.889, respectively, indicating good reliability of our self-developed questionnaire content.

### 3.2. Research Objects

The study group comprised nurses from different units of a regional hospital in Taipei City, Taiwan. A total of 220 questionnaires were assessed, and then the ratio of observations to independent variables was calculated at a ratio of 10:1 and at a 50% recovery rate by convenience sampling.

### 3.3. Research Process

The researcher distributed the questionnaires to each ward with instructions for filling out and obtaining the subjects' consent form. The consent form takes about 30 min to fill out. The questionnaire is anonymous. The research subjects put the questionnaire in the data bag and sealed it after completing the filling out before the agreed date, and it was stored in a fixed place in the unit before being collected by the researcher.

### 3.4. Statistical Analysis of Data

We recoded the questionnaire data, and the results were compiled with SPSS for Windows version 20.0 statistical software and analyzed by descriptive and inferential statistics.

### 3.5. Test Results

A total of 220 questionnaires were assessed, and 218 were returned, with a recovery rate of 99.1%. There were 203 valid questionnaires, accounting for 93.1%. The results of compilation and analysis by SPSS for Windows version 20.0 statistical software are as follows:

### 3.6. Socio-Demographic Characteristics

The age of nurses in the study group ranged between 20 and 47 years, with a mean age of 28.67 years (SD = 6.33). For gender, 100% of the valid samples are women. For educational status, 111 persons (54.7%) had a university degree, and 92 persons (45.3%) had a college degree. The number of years of nursing career showed a mean of 6.258 years (SD = 5.819). The telephone follow-up interview experience of $n = 35$ people (17.3%) had telephone follow-up interview experience at work, and $n = 167$ people (82.3%) had no such experience. A total of $n = 145$ people (72.1%) had experience in family-in-call consultation at work, and $n = 56$ people (27.9%) had no such experience. The information for telehealth contact showed that $n = 95$ people (46.8%) had experienced telehealth information before participating in this study, and $n = 108$ people (53.2%) were inexperienced. Before participating in this study, $n = 95$ people (46.8%) had experience in telehealth information and $n = 108$ people (53.2%) were inexperienced. The mean years of using the medical information system was 2.975 years (SD = 3.509). For experience in using telehealth systems, $n = 33$ nurses (16.3%) had experience using telehealth systems, and $n = 169$ nurses (86.7%) had no experience (see Table 1).

**Table 1.** Socio-demographic characteristics of the research sample ($n = 203$).

| Variable | Category | Number | Percentage (%) |
|---|---|---|---|
| Age [+] | | 199 | 100.0 |
| Sex | Female | 203 | 100.0 |
| Education Level | College | 92 | 45.3 |
| | Bachelor | 111 | 54.7 |
| Years of nursing career [+] | | 201 | 100.0 |
| Have telephone follow-up interview experience | Yes | 35 | 17.3 |
| | No | 167 | 82.7 |
| Experience in responding to phone inquiry from patients' family | Yes | 145 | 72.1 |
| | No | 56 | 27.9 |
| Telenursing knowledge | Yes | 95 | 46.8 |
| Years of using medical information system | No | 108 | 53.2 |
| | | 197 | 100.0 |
| Experience in using telenursing system | Yes | 33 | 16.3 |
| | No | 169 | 83.7 |

[+] Missing Value.

### 3.7. Socio-Demographic Characteristics and Differences in Research Variables

The socio-demographic characteristics of the sample, the seven extracted factors, three independent variables, and dependent variables were tested by Independent-Sample t-Test and Pearson's product-moment correlation. Analysis of differences was conducted. The results are shown in Table 2.

As shown in Table 2, compared to nurses without telenursing knowledge, the behavior of nurses with telenursing knowledge was positively influenced by perceived usefulness ($t = 2.519$; $p < 0.05$), subjective norms ($t = 2.082$; $p < 0.05$), supervisor influence ($t = 2.211$; $p < 0.05$), perceived behavior control ($t = 2.892$; $p < 0.01$), self-efficacy ($t = 3.135$; $p < 0.01$), and behavioral intention ($t = 2.762$; $p < 0.01$). Among nurses who had experience in using telehealth systems, behavioral intentions had a more positive influence toward behavior ($t = 2.762$; $p < 0.01$) than among those without prior experience. Peer influence had less influence on behavior for university graduates compared to college graduates ($t = 2.105$; $p < 0.05$).

**Table 2.** Differences between study population characteristics and research variables.

| Variable | Education Level | | | Have Telephone Follow-Up Interview Experience | | | Experience in Responding to Phone Inquiry from Patients' Family | | | Knowledge in Telenursing | | | Experience in Using Telenursing System | | |
|---|---|---|---|---|---|---|---|---|---|---|---|---|---|---|---|
| | Level | M ± SD | t | Level | M ± SD | t | Level | M ± SD | t | Level | M ± SD | t | Level | M ± SD | t |
| Attitude | College | 78.08 ± 8.72 | 0.526 | No | 77.49 ± 9.32 | −1.069 | No | 77.13 ± 11.41 | −0.558 | No | 76.72 ± 10.23 | −1.635 | No | 77.59 ± 9.11 | −0.515 |
| | Bachelor | 77.4 ± 9.46 | | Yes | 79.29 ± 7.41 | | Yes | 78.06 ± 7.97 | | Yes | 78.82 ± 7.57 | | Yes | 78.49 ± 9.34 | |
| Perceived usefulness | College | 41.25 ± 4.8 | −0.2 | No | 41.27 ± 5.19 | −0.788 | No | 41.13 ± 6.21 | −0.49 | No | 40.5 ± 5.67 | −2.519 * | No | 41.26 ± 5.15 | −0.58 |
| | Bachelor | 41.4 ± 5.31 | | Yes | 42 ± 4.08 | | Yes | 41.52 ± 4.51 | | Yes | 42.27 ± 4.14 | | Yes | 41.82 ± 4.75 | |
| Perceived ease of use | College | 24.92 ± 2.96 | 0.936 | No | 24.56 ± 3.08 | −1.655 | No | 24.44 ± 3.59 | −0.774 | No | 24.59 ± 3.37 | −0.569 | No | 24.68 ± 3 | −0.399 |
| | Bachelor | 24.52 ± 3.07 | | Yes | 25.49 ± 2.63 | | Yes | 24.81 ± 2.78 | | Yes | 24.83 ± 2.6 | | Yes | 24.91 ± 3.21 | |
| Compatibility | College | 11.9 ± 1.83 | 1.447 | No | 11.67 ± 2.04 | −0.35 | No | 11.56 ± 2.42 | −0.538 | No | 11.64 ± 2.06 | −0.228 | No | 11.65 ± 2.01 | −0.265 |
| | Bachelor | 11.48 ± 2.23 | | Yes | 11.8 ± 2.07 | | Yes | 11.74 ± 1.89 | | Yes | 11.7 ± 2.08 | | Yes | 11.76 ± 2.37 | |
| Subjective norms | College | 15.77 ± 2.42 | 1.344 | No | 15.47 ± 2.45 | −0.842 | No | 15.46 ± 2.9 | −0.282 | No | 15.19 ± 2.58 | −2.082 * | No | 15.41 ± 2.35 | −1.396 |
| | Bachelor | 15.31 ± 2.42 | | Yes | 15.85 ± 2.25 | | Yes | 15.58 ± 2.22 | | Yes | 15.89 ± 2.18 | | Yes | 16.06 ± 2.76 | |
| Supervisor influence | College | 7.87 ± 1.3 | 0.264 | No | 7.78 ± 1.32 | −1.501 | No | 7.78 ± 1.5 | −0.448 | No | 7.65 ± 1.33 | −2.211 * | No | 7.8 ± 1.27 | −1.068 |
| | Bachelor | 7.82 ± 1.29 | | Yes | 8.14 ± 1.14 | | Yes | 7.88 ± 1.21 | | Yes | 8.05 ± 1.22 | | Yes | 8.06 ± 1.39 | |
| Peer influence | College | 7.9 ± 1.29 | 2.105 * | No | 7.69 ± 1.32 | −0.314 | No | 7.67 ± 1.48 | −0.17 | | 7.53 ± 1.38 | −1.778 | No | 7.61 ± 1.3 | −1.763 |
| | Bachelor | 7.51 ± 1.33 | | Yes | 7.77 ± 1.3 | | Yes | 7.71 ± 1.26 | | Yes | 7.86 ± 1.24 | | Yes | 8.06 ± 1.46 | |
| Perceived behavior control | College | 30.95 ± 3.72 | 0.981 | No | 30.56 ± 3.88 | −0.772 | No | 30.34 ± 4.81 | −0.596 | No | 29.96 ± 4.23 | −2.892 ** | No | 30.54 ± 3.73 | −0.916 |
| | Bachelor | 30.42 ± 3.88 | | Yes | 31.12 ± 3.51 | | Yes | 30.76 ± 3.37 | | Yes | 31.46 ± 3.09 | | Yes | 31.22 ± 4.27 | |
| Self-efficacy | College | 20.16 ± 2.54 | 0.8 | No | 19.89 ± 2.54 | −1.142 | No | 19.68 ± 3.25 | −0.872 | No | 19.5 ± 2.88 | −3.135 ** | No | 19.99 ± 2.6 | −0.138 |
| | Bachelor | 19.87 ± 2.59 | | Yes | 20.44 ± 2.63 | | Yes | 20.09 ± 2.24 | | Yes | 20.59 ± 2.01 | | Yes | 20.06 ± 2.49 | |
| Facilitating conditions | College | 10.78 ± 1.95 | 0.849 | No | 10.67 ± 2 | −0.016 | No | 10.66 ± 2.3 | −0.019 | No | 10.46 ± 2.14 | −1.494 | No | 10.55 ± 1.91 | −1.596 |
| | Bachelor | 10.55 ± 2 | | Yes | 10.68 ± 1.79 | | Yes | 10.67 ± 1.83 | | Yes | 10.87 ± 1.76 | | Yes | 11.16 ± 2.26 | |
| Behavioral intention | College | 24 ± 4.35 | 0.152 | No | 23.83 ± 4.65 | −1.035 | No | 23.82 ± 5.68 | −0.267 | No | 23.12 ± 4.71 | −2.726 ** | No | 23.64 ± 4.45 | −2.252 * |
| | Bachelor | 23.9 ± 4.87 | | Yes | 24.71 ± 4.46 | | Yes | 24.01 ± 4.17 | | Yes | 24.87 ± 4.39 | | Yes | 25.61 ± 5.27 | |

Note: Independent *t*-test analysis; * *p* < 0.05 ** *p* < 0.01.

Pearson's product-moment correlation was used to evaluate the correlation of variables by age, years of nursing career, and years of experience in using the health care information system. The results showed no statistically significant differences (Table 3).

**Table 3.** Differences in characteristics of the study sample population and research variables.

| Variable | M ± SD | Coefficient of Correlation Analysis (r) | | |
| --- | --- | --- | --- | --- |
| | | Age | Years of Nursing Career | Year of Using Medical Information System |
| Age | 28.68 ± 6.32 | | | |
| Nursing career (yr) | 6.26 ± 5.82 | | | |
| Using medical info system (yr) | 2.98 ± 3.51 | | | |
| Attitude | 77.7 ± 9.12 | 0.025 | −0.036 | −0.004 |
| Perceived usefulness | 41.33 ± 5.08 | 0.021 | −0.037 | 0.01 |
| Perceived ease of use | 24.7 ± 3.02 | 0.011 | −0.055 | −0.03 |
| Compatibility | 11.67 ± 2.07 | 0.041 | 0.01 | 0.004 |
| Subjective norms | 15.52 ± 2.42 | −0.031 | −0.008 | 0.057 |
| Supervisor influence | 7.84 ± 1.29 | −0.003 | −0.008 | 0.05 |
| Peer influence | 7.69 ± 1.33 | −0.038 | −0.012 | 0.053 |
| Perceptual behavior control | 30.66 ± 3.81 | −0.008 | 0.012 | 0.085 |
| Self-efficacy | 20.01 ± 2.57 | 0.071 | 0.068 | 0.111 |
| Facilitating conditions | 10.65 ± 1.98 | −0.107 | −0.063 | 0.021 |
| Behavioral intention | 23.95 ± 4.63 | −0.048 | −0.064 | −0.023 |

Note: Analyze with Pearson's correlation.

*3.8. The Correlation between Research Variables*

Pearson's product-moment correlation was used to compare perceived usefulness, perceived ease of use, compatibility, supervisory influence, peer influence, self-efficacy, support conditions, attitude, subjective norms, and conscious behavior control. The behavioral intentions were tested for correlation to conduct further multiple regression analysis of the variables whose correlations reach statistically significant differences (multiple regression analysis).

The correlation matrix results show that all variables and behavioral intentions were positively correlated, and all with statistical significance ($p < 0.001$).

In terms of the strength of relevance, moderately related to behavioral intentions are supervisory influence, facilitating conditions, subjective norms, and perceived behavior control, among which facilitating conditions and behavioral intention were the strongest ($r = 0.501$, $p < 0.001$), followed by perceived behavior control and behavioral intention ($r = 0.468$, $p < 0.001$), and between subjective norms and behavioral intention ($r = 0.430$, $p < 0.001$). The influence of supervisors and behavioral intention were second ($r = 0.414$, $p < 0.001$). Lower-level correlations with behavioral intention were perceived usefulness, perceived ease of use, compatibility, peer influence, self-efficacy, and attitudes. Among these, the relationship between peer influence and behavioral intention was the strongest ($r = 0.394$, $p < 0.001$), followed by the relationship between perceived usefulness and behavioral intention ($r = 0.375$, $p < 0.001$), between attitude and behavioral intention ($r = 0.372$, $p < 0.001$), between self-efficacy and behavioral intention ($r = 0.313$, $p < 0.001$), between compatibility and behavioral intention ($r = 0.298$, $p < 0.001$), and between perceived ease of use and behavioral intention ($r = 0.287$, $p < 0.001$) (see Table 4).

**Table 4.** Correlation matrix of each variable and behavioral intention.

| Variable | Perceived Usefulness | Perceived Ease of Use | Compatibility | Supervisor Influence | Peer Influence | Self-Efficacy | Facilitating Conditions | Attitude | Subjective Norms | Perceived Behavior Control | Behavioral Intention |
|---|---|---|---|---|---|---|---|---|---|---|---|
| Perceived usefulness | 1 | | | | | | | | | | |
| Perceived ease of use | 0.745 *** | 1 | | | | | | | | | |
| Compatibility | 0.618 *** | 0.633 *** | 1 | | | | | | | | |
| Supervisor influence | 0.687 *** | 0.554 *** | 0.453 *** | 1 | | | | | | | |
| Peer influence | 0.557 *** | 0.469 *** | 0.406 *** | 0.720 *** | 1 | | | | | | |
| Self-efficacy | 0.578 *** | 0.528 *** | 0.373 *** | 0.531 *** | 0.441 *** | 1 | | | | | |
| Facilitating conditions | 0.395 *** | 0.330 *** | 0.420 *** | 0.437 *** | 0.515 *** | 0.396 *** | 1 | | | | |
| Attitude | 0.946 *** | 0.893 *** | 0.781 *** | 0.673 *** | 0.560 *** | 0.584 *** | 0.426 *** | 1 | | | |
| Subjective norms | 0.667 *** | 0.547 *** | 0.458 *** | 0.925 *** | 0.930 *** | 0.523 *** | 0.514 *** | 0.660 *** | 1 | | |
| Perceived behavior control | 0.597 *** | 0.529 *** | 0.472 *** | 0.587 *** | 0.567 *** | 0.879 *** | 0.786 *** | 0.618 *** | 0.622 *** | 1 | |
| Behavioral intention | 0.375 *** | 0.287 *** | 0.298 *** | 0.414 *** | 0.394 *** | 0.313 *** | 0.501 *** | 0.372 *** | 0.430 *** | 0.468 *** | 1 |

Note: Analyzed with Pearson's product-moment correlation. *** $p < 0.001$.

### 3.9. Predictors of Behavioral Intention

The correlation between various variables (perceived usefulness, perceived ease of use, compatibility, supervisor influence, peer influence, self-efficacy, facilitating conditions, and sociodemographic characteristics) and behavioral intentions was statistically significant. To identify predictive factors, stepwise multiple regression analysis was applied—the independent variable (predictive variable) was experience with the telehealth system, and the dependent variable (standard variable) was behavioral intention.

The results of multiple regression analysis with behavioral intentions regressed on facilitating conditions and supervisory influence showed that "Behavioral intention = 7.543 + 0.394 × Facilitating conditions + 0.232 × Supervisory influence" (Table 5). The determinant coefficient ($R^2$) of the two predictive variables (facilitating conditions and supervisor influence) for behavioral intention and the standard variable was 0.293, and $R^2$ after adjustment was 0.286. The F value of the statistical significance test of the variance was 40.394 ($p < 0.001$). These two predictive variables jointly explained 28.6% of the variance for the variable "behavioral intention", and the overall explained variance reached statistical significance level. Among the predictive variables, facilitating condition had more significant influence on behavioral intention ($\beta = 0.394$), followed by supervisor influence ($\beta = 0.232$). The t-values for the statistical significance test of the regression coefficients were 5.817 ($p < 0.001$) for facilitating condition and 3.431 ($p < 0.01$) for supervisor influence. Both significantly predict behavioral intention toward practicing telenursing.

**Table 5.** Summary table of multiple regression analysis of behavioral intention predictors.

| Predictor | B | $\beta$ | t | F | $R^2$ | Adjusted $R^2$ |
|---|---|---|---|---|---|---|
| Criterion = Behavioral Intention | | | | | | |
| Constant | 7.543 | | 3.976 *** | | | |
| Facilitating conditions | 0.923 | 0.394 | 5.817 *** | | | |
| Supervisor influence | 0.832 | 0.232 | 3.431 ** | 40.394 *** | 0.293 | 0.286 |

Dependent variable: Behavioral intention; R = 0.541, ** $p < 0.01$, *** $p < 0.001$.

To clarify collinearity, the variance inflation factors (VIF) were examined, and the relative weight and relative importance were also calculated for all variables and three sub-domains (attitude, subjective norms, and PBC). The relative importance proportions were 46.68%, 33.34%, and 19.98% for sub-domain PBC, subjective norms, and attitude, respectively. Among all variables, facilitating conditions and supervisor influence had the highest relative importance (RW = 0.135 and 0.046) (Supplementary Material Table S2).

In "facilitating conditions", stronger behavioral intention for practicing telenursing was associated with higher scores in the following questions: "the nursing staff work organization/institute has offered telehealth-related on-the-job education courses"; "the work organization provides education and training on telehealth-related knowledge and information systems"; and "user responses to the information system will soon get feedback from the information unit." In terms of supervisor influence, the higher the scores of "receiving support from supervisors" and "understanding that telehealth is the future trend of healthcare policies", the stronger the intention to practice telenursing.

## 4. Discussion

### 4.1. The Relationship between Attitude and Behavioral Intention

In this study, nurses' attitudes towards telenursing correlated positively with their intention to participate in telehealth-related work. Nurses who already had experienced telehealth information had a more positive attitude towards telenursing than those who had not encountered telehealth information. These observations are in line with nurses' behavioral intention using Web 2.0 by Lau scholars, who observed a positive correlation between attitude and behavioral purposes [34].

Similarly, a study of telehealth cognition of college nursing students showed that 67% of individuals were willing to participate in telehealth services, and 69.49% of people

agreed that knowledge of telehealth should be included in school courses [35]. Therefore, prior experience in telehealth education creates familiarity and understanding of the terms of cognition and knowledge, and being informed of telehealth increases acceptance, which may increase the degree of nursing staff involvement.

### 4.2. The Relationship between Subjective Norms and Behavioral Intentions

In the present study, subjective norms included the influence of peers and supervisors. Nurses' subjective norms for telenursing correlated positively with their behavioral intentions to practice telenursing. Similarly, Lau et al. found that supervisory influence correlated positively with hospital staff's behavioral intention [34]. This result also shows that the leadership direction of the organization's peers or supervisors affected behavioral intention [22]. This also suggests that if the organization's development direction is toward technology solutions, organizing teams to develop telehealth services together may encourage the participation of nurses in telenursing.

### 4.3. The Relationship between Perceived Behavior Control and Behavioral Intention

In the present study, perceived behavior includes self-efficacy and facilitating conditions. The results of the present study showed that good self-efficacy and facilitating conditions correlated positively with nurses' intention to participate in telenursing. Similarly, in another study that used the DTPB to investigate intentions, five factors were found to significantly influence physicians' intentions to use electronic medical record (EMR) exchange, demonstrating that the DTPB model effectively predicts intentions to participate [36]. Similarly, a study by Gonen et al. (2014) [37] on students' attitudes about computer use found that students lacking a sense of threat, self-use experience and skills with computers were associated with better attitudes toward using computers. Correspondingly, having relevant knowledge, familiarity with technology, supportive facilitating conditions, and increased self-capacity may enhance behavioral intentions to participate in telenursing.

### 4.4. Predictors of Intention to Participate in Telehealth

The results of the present study show that "facilitating conditions" and "supervisory influence" were two statistically supported predictors of nursing staff's intention to participate in telenursing. Getting support from the supervisor increases nurses' willingness to participate in telenursing, suggesting that policy support of the superiors, hospital administrators or institution support are important. This result is consistent with those of Tourangeau et al. (2010) [38] that good relationships between nursing staff and supervisors come from having the support of supervisors and organizations.

Facilitating conditions include quick response to questions or concerns about using telehealth information systems. Health care organizations must provide knowledge and training on telenursing and telehealth systems by creating an environment that offers strong administrative support and educational resources, which will provide a consistent theoretical framework and ultimately improve nurses' intention to participate in telenursing [25].

The main obstacle for nurses in telenursing participation is lack of knowledge and skills [39]. Nurses will sometimes resist for fear of learning new technologies in telehealth development. However, proper training and administrative support may reduce operational resistance [16]. Training and education in telehealth and information systems courses have helped in increased technology facilitating and resource facilitating conditions, improving participation in telenursing [40]. Studies have confirmed that compatibility, peer influence, perceived usefulness, perceived ease of use, self-efficacy, and other factors which were moderately related to behavioral intentions contribute to making behavioral decisions [41]. Jourdain and Chênevert (2010) [42] demonstrated that multiple strategies are necessary, including reducing work requirements and increasing available work resources; thus, adjusting the work and roles of nurses can sufficiently reduce work overload and

improve the value of their work. The results of the present study indicated that nurses who have experience in telehealth have significantly greater perceived usefulness than those who are inexperienced with telehealth, which correlated positively with attitudes and behavioral intentions. Experience is shown to have an influence on the display of behavioral intention associated with more positive attitude [25].

The ages of the nurses correlated positively with telenursing attitudes but correlated negatively with behavioral intentions to participate in telenursing. Although this may be due to positive attitudes, other considerations and complications, such as salary, working environment and position, may be involved. Multiple complex factors such as improving family lifestyle and working conditions are associated with factors that trigger nurses to seek new jobs [43].

### 4.5. Limitations

The present study has several limitations, including that findings and implications drawn from this study cannot be generalized because the study participants only included nursing staff from a single center. Furthermore, all respondents were female, and the effect of sex on behavioral intentions and technology acceptance should be investigated in future studies. Additionally, barriers to the use of technology were not addressed.

Lastly, we did not measure actual behavior in this study to determine whether the theoretical results demonstrated actual trends in nursing behavior. Cross-sectional design was implemented in this study—one reason for not using a longitudinal design was that telehealth is a relatively new domain for nurses in Taiwan, and we do not yet know the potential influencing factor nor how long it will take for these potential covariates to exert an effect on behavioral intention for participating in telenursing. Nevertheless, our finding provided initial evidence to indicate important factors (i.e., facilitating conditions and encouragement from nursing supervisors) related to nurses' intention to practice telenursing. As more information becomes available, subsequent prospective studies allowing the establishment of causation and mediators should be designed.

### 5. Conclusions and Suggestions

The attitudes, subjective norms, and perceived behavioral control of nurses toward telenursing correlated positively with the behavioral intentions of nurses to participate in telenursing. Facilitating conditions and supervisor influence are the leading predictors of nurses' behavioral intentions.

In the present study, the application of the DTPB model helped to identify influencing factors for the acceptance of the telenursing concept by nurses, which will further support the expansion of telehealth through increased adoption. The results of this study suggested that telenursing participation can be encouraged at the educational level by incorporating it into school courses and seminars or webinars, as well as promoting the concept through health care organizations. Such promotion may help to increase the visibility of telehealth advantages and increase the willingness of nurses and the public to accept telenursing.

**Supplementary Materials:** The following are available online at https://www.mdpi.com/article/10.3390/informatics8020035/s1, Table S1: The questionnaire in this study, Table S2: Summary of a traditional relative weight.

**Author Contributions:** M.-Y.C.: study concepts; study design; definition of intellectual content; manuscript preparation; manuscript editing; manuscript review. F.-L.K.: study design; definition of intellectual content; literature research; clinical studies; data acquisition; data analysis; statistical analysis; manuscript preparation. C.-C.L. and T.-R.L.: literature research; statistical analysis; manuscript editing; manuscript review. T.-Y.L.: guarantor of integrity of the entire study; study concepts; study design; definition of intellectual content; literature research; clinical studies; experimental studies; data acquisition; data analysis; statistical analysis; manuscript preparation; manuscript editing; manuscript review. All authors have read and agreed to the published version of the manuscript.

**Funding:** This study was supported by research funding from Cheng Hsin General Hospital (CHGH105-36).

**Institutional Review Board Statement:** This study was approved by the Institutional Review Board of the hospital (No. 487-104-23).

**Informed Consent Statement:** Written informed consent was obtained from all subjects included in this study.

**Data Availability Statement:** The data and materials used in this study are available from the corresponding author on reasonable request.

**Acknowledgments:** The authors wish to thank Cheng Hsin General Hospital for funding the research (CHGH105-36) and research sites.

**Conflicts of Interest:** All authors have no conflict of interest to declare.

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
