# Peer review of "The Intention and Influence Factors of Nurses’ Participation in Telenursing"

_informatics, doi:10.3390/informatics8020035_

Round 1
Reviewer 1 Report
Thank you very much for inveting me to review the revised version of this manuscript. It there are now only some minor issues that need to be revised before a publication is possible.
1. Formal recommendation: Space characters
In the manuscript and the supplement space characters between words, after commas, after sentences (end -of-record) are often missing (see my eralier comment 6). Please check all documents thoroughly.
In the supplement there is also a typo "Being supported by supervisors will enhance ... telenursing".
2. p.6., l.6: "In the present study, the construct measures shown in Figure 1 were"
3. p.7., l.5 "with a mean age of 28.67 years (SD = 6.33).
4. p.7, l.11 "had no such experience. A total..."
5. Limitations (4.5): Please discuss using implications of testing a causal model (Fig 1) with cross-sectional data.
You might also use some arguments from this paper why a cross-sectional design might be warranted to answer your research questions.
Spector, P. E. (2019). Do not cross me: Optimizing the use of cross-sectional designs. Journal of Business and Psychology, 34(2), 125–137. https://doi.org/10.1007/s10869-018-09613-8
Author Response
Section Managing Editor
Informatics Editorial Office
Re: Manuscript ID: informatics-1209947
Dear Editor,
We are resubmitting our manuscript entitled “The Intention and Influence Factors of Nurses’ Participation to Telenursing” with the requested revisions by the reviewers. We thank you and the reviewers for your time and constructive comments. Our point-by-point responses addressing the reviewers’ comments are as follows.
As instructed,revision carried out according to the reviewers’ comments were highlighted and made easily readable for the editors and reviewers. (Please note, for ease of reading, minor grammar changes, typos and character spacing issues were not highlighted). We hope you will find the revised manuscript now suitable for publication in Informatics. We look forward to your reply.
Sincerely,
Correspondence to: Tso-Ying Lee
Point-by-point response to reviewers
Reviewer #1
Comment: Thank you very much for inviting me to review the revised version of this manuscript. It there are now only some minor issues that need to be revised before a publication is possible.
1. Formal recommendation: Space characters
In the manuscript and the supplement space characters between words, after commas, after sentences (end -of-record) are often missing (see my earlier comment 6). Please check all documents thoroughly.
In the supplement there is also a typo "Being supported by supervisors will enhance ... telenursing".
2. p.6., l.6: "In the present study, the construct measures shown in Figure 1 were"
3. p.7., l.5 "with a mean age of 28.67 years (SD = 6.33).
4. p.7, l.11 "had no such experience. A total..."
Response to Comment 1-4: Thank you for reviewing our manuscript, we have thoroughly checked the manuscript for spacing between characters, as well as typos and grammar. The type mentioned above were revised, please refer to the yellow highlights in the manuscript. For ease of reading, however, other formatting and grammar issues were not highlighted in the revised manuscript.
5. Limitations (4.5): Please discuss using implications of testing a causal model (Fig 1) with cross-sectional data.
You might also use some arguments from this paper why a cross-sectional design might be warranted to answer your research questions.
Spector, P. E. (2019). Do not cross me: Optimizing the use of cross-sectional designs. Journal of Business and Psychology, 34(2), 125–137. https://doi.org/10.1007/s10869-018-09613-8
Response to Comment 5: Thank you for suggesting a very useful and informative reference. We have added a discussion on use of cross-sectional design to the limitation section, as follows:
“Cross-sectional design was implemented in this study, one reason for not using a longitudinal design was that telehealth is a relatively new domain for nurses in Taiwan, we do not yet know the potential influencing factor nor how long for these potential covariates to exert an effect on behavioral intention for participating in telenursing. Nevertheless, our finding provided initial evidence to indicate important factors (i.e., facilitating conditions and encouragement from nursing supervisors) related to nurses’ intention to practice telenursing. As more information becomes available, subsequent prospective studies al-lowing establishment of causation and mediators should be designed”.
Reviewer #2
Comment: You did a great job applying the decomposed theory of planned behavior. The paper is very sound and well written. My one criticism is that the results do not offer impactful findings.
Response: Thank you for reviewing our manuscript.
Telehealth is a relatively new domain for nurses in Taiwan, it was not yet clear of the potential influencing factor nor how long for these potential covariates to exert an effect on behavioral intention for participating in telenursing. Nevertheless, our current finding provided initial evidence to indicate some important factors (i.e., facilitating conditions and encouragement from nursing supervisors) related to nurses’ intention to practice telenursing. We agree that as more information becomes available, subsequent prospective studies allowing establishment of causation and mediators should be designed to obtain more impactful results.

Reviewer 2 Report
You did a great job applying the decomposed theory of planned behavior. The paper is very sound and well written. My one criticism is that the results do not offer impactful findings.
Author Response
Section Managing Editor
Informatics Editorial Office
Re: Manuscript ID: informatics-1209947
Dear Editor,
We are resubmitting our manuscript entitled “The Intention and Influence Factors of Nurses’ Participation to Telenursing” with the requested revisions by the reviewers. We thank you and the reviewers for your time and constructive comments. Our point-by-point responses addressing the reviewers’ comments are as follows.
As instructed,revision carried out according to the reviewers’ comments were highlighted and made easily readable for the editors and reviewers. (Please note, for ease of reading, minor grammar changes, typos and character spacing issues were not highlighted). We hope you will find the revised manuscript now suitable for publication in Informatics. We look forward to your reply.
Sincerely,
Correspondence to: Tso-Ying Lee
Point-by-point response to reviewers
Reviewer #1
Comment: Thank you very much for inviting me to review the revised version of this manuscript. It there are now only some minor issues that need to be revised before a publication is possible.
1. Formal recommendation: Space characters
In the manuscript and the supplement space characters between words, after commas, after sentences (end -of-record) are often missing (see my earlier comment 6). Please check all documents thoroughly.
In the supplement there is also a typo "Being supported by supervisors will enhance ... telenursing".
2. p.6., l.6: "In the present study, the construct measures shown in Figure 1 were"
3. p.7., l.5 "with a mean age of 28.67 years (SD = 6.33).
4. p.7, l.11 "had no such experience. A total..."
Response to Comment 1-4: Thank you for reviewing our manuscript, we have thoroughly checked the manuscript for spacing between characters, as well as typos and grammar. The type mentioned above were revised, please refer to the yellow highlights in the manuscript. For ease of reading, however, other formatting and grammar issues were not highlighted in the revised manuscript.
5. Limitations (4.5): Please discuss using implications of testing a causal model (Fig 1) with cross-sectional data.
You might also use some arguments from this paper why a cross-sectional design might be warranted to answer your research questions.
Spector, P. E. (2019). Do not cross me: Optimizing the use of cross-sectional designs. Journal of Business and Psychology, 34(2), 125–137. https://doi.org/10.1007/s10869-018-09613-8
Response to Comment 5: Thank you for suggesting a very useful and informative reference. We have added a discussion on use of cross-sectional design to the limitation section, as follows:
“Cross-sectional design was implemented in this study, one reason for not using a longitudinal design was that telehealth is a relatively new domain for nurses in Taiwan, we do not yet know the potential influencing factor nor how long for these potential covariates to exert an effect on behavioral intention for participating in telenursing. Nevertheless, our finding provided initial evidence to indicate important factors (i.e., facilitating conditions and encouragement from nursing supervisors) related to nurses’ intention to practice telenursing. As more information becomes available, subsequent prospective studies al-lowing establishment of causation and mediators should be designed”.
Reviewer #2
Comment: You did a great job applying the decomposed theory of planned behavior. The paper is very sound and well written. My one criticism is that the results do not offer impactful findings.
Response: Thank you for reviewing our manuscript.
Telehealth is a relatively new domain for nurses in Taiwan, it was not yet clear of the potential influencing factor nor how long for these potential covariates to exert an effect on behavioral intention for participating in telenursing. Nevertheless, our current finding provided initial evidence to indicate some important factors (i.e., facilitating conditions and encouragement from nursing supervisors) related to nurses’ intention to practice telenursing. We agree that as more information becomes available, subsequent prospective studies allowing establishment of causation and mediators should be designed to obtain more impactful results.

This manuscript is a resubmission of an earlier submission. The following is a list of the peer review reports and author responses from that submission.
Round 1
Reviewer 1 Report
Thank you very much for inviting me to review this study. I have the following recommendations for further improvement in order to be published.
1. Abstract
Please add the following information:
- What is the study goal? (Our study aims to ...)
- Study design: cross-sectional survey
- Sample information: 203 nurses from a regional hospital in Taipei City (China)
- What are the key results from you analyses
2. Theory
2.1 Introduction
Please consider to give short review on the results of the literatur review and add to the last paragraph of the introdcution what your study contributes to existing literature when answering the main research questions.
2.2 Review
2.2.1 Figure 1: There is some information missing in the box on behaviroal intention; please consider to revice the box on subjective norm (missing spaces)
2.2.2 Evidence on the TPB
I am missing a bit the published literature on the evidence of TBP and the technology acceptance model for nurses attitudes towards digital technologies and, more specifically, telecare.
Please consider at least the following papers:
King WR, He J. A meta-analysis of the technology acceptance
model. Information & Management 2006; 43(6): 740-755..
Schlicht, L., Wendsche. J., Lehrke, L., Melzer, M., & Rösler, U. (2020). Nurses’ attitudes towards occupational transformation processes brought about by digital care technologies: Results from two cross-sectional studies. Current Directions in Biomedical Engineering, 6(3), 20203160. https://doi.org/10.1515/cdbme-2020-3160
The paper of Schlicht et al. will give you also some information on telecare acceptance and potential drivers in two German nurse samples (check alos the reference list for other studies on that topic).
2.3 Research questions and hypotheses
Under 2.5 please add at least research questions or better direceted hypotheses that fit with your analyses. You have the model in Figure 1 which is helpful here.
Otherwise your analyses come a bit arbitrary and the reader those not really know what your are doing. Thus, such a section will give your paper more structure. Please note that this needs to give response to the RQs or Hypotheses in the results section.
2.4 The research model in Figure 1
Is it possible to integrate the other relavnt variables such as education and telecare experince in your research model. That would be a nice integration but also helps the reade to get a fast overview on the study variables.
3. Methods
3.1 Scales
Where do your scales and questions come from? Are these established scales or self-developed? This has to be reported (please add a reference for each scale, the number of items, a sample item) for each constructz and variable. Otherwise replication is not possible. If it is all self-developed please add all scales and items in an appendix. What was the response format of the Likert scales?
3.2 Construct Validity
I would like to encourage you to add further information on construct validity. This could be the results of an exploratory or better confirmatory factor analysis showing that all TPB variables are independent. This is also necessary since the intercorrelations in your study are extremly high.
4. Results
4.1 p. 9. first paragraph: I do not really understand what you are writing here. Do you mean that experienced nurses report higher usefulness, and so on? Please revise this section.
4.2 Table 3: This is hard to read because you report so many decimals. From my persepective 2 decimals are enough.
4.3 Variance inflation
Since your varaibles are all highly correlated you should report variance inflation factors (VIF) or, the better apporach, conduct a relative importance or relative weights analysis.
You find an online tool for such analyses here:
Tonidandel, S., LeBreton, J.M. RWA Web: A Free, Comprehensive, Web-Based, and User-Friendly Tool for Relative Weight Analyses. J Bus Psychol 30, 207–216 (2015). https://doi.org/10.1007/s10869-014-9351-z
The regression equation can be deleted (above table 5). Since you have some many tables I would also like to encoruage you tor report the results of Table 5 in the text: "results of a multiple regression analysis with behavior intentions regressed on ... showed that ...". Why are the other predictors not integrated in that analysis such as usefulness, attioudes and so on. Form my perspecte the best way to test the model with ablockwise regression is:
Dependent Variable: Behavior Intentions
- Controls
- Attitudes, subjective norms and PBC (it is also to include the specific variables for the three factors blockwise)
4.4 The paper of Schlicht et al found also an interaction effect of knwoledge and usability on attitudes for some digital technologies (but not for telecare). Is it possible to check this in a sublementary analysis.
5. Discussion
5.1 The are some grammatical errors here. Please conduct proof reading.
5.2 "The results of this study are similar to the results of a survey on obesity caregivers' exercise intentions." I would recommend to report on meta-analytical results on the TBP or on results regrading nurses attitudes towards digital technologies. Attitudes towards excercise bevaior are something different. This literature does not fit with your study (results).
5.3 What are the limitations of your study? Please add a section on this.
6. Other Comments
- Please check the full manuscript for missing spaces between words (e.g. p.2., l 15: "TheCOVID-19 pandemichad" should be "The COVID-19 pandemic had.."; there are many of theses typos in the manuscript that need a throughly revision (check also the references; e.g. RogersM.Everett. Diffusion of Innovations: Third Edition. 1983.)
Author Response
Dear Editor,
We are resubmitting our manuscript entitled “The Intention and Influence Factors of Nurses’ Participation to Telenursing” with the requested revisions by the reviewers. We thank you and the reviewers for your time and constructive comments. Our point-by-point responses addressing the reviewers’ comments are as follows.
We hope you will find the revised manuscript is now substantially improved and suitable for publication in Informatics. We look forward to your reply.
Sincerely,
Correspondence to:
Tso-Ying Lee, Associate Professor, Ph.D. (the corresponding author)
Nursing Department of Cheng Hsin General Hospital
3 F, No. 45, Cheng Hsin St., Beitou Dist., Taipei City 105, Taiwan.
E-mail: tsoyinglee@gmail.com
Telephonenumber:886-975359825
Fax number:886-2-28264574
Point-by-point response to reviewers
Reviewer #1
Abstract
Please add the following information:
What is the study goal? (Our study aims to ...)
Study design: cross-sectional survey
Sample information: 203 nurses from a regional hospital in Taipei City (China)
What are the key results from your analyses?
Author response: The Abstract has been revised accordingly. Please see the revised abstract section.
- Theory
2.1 Introduction
Please consider to give short review on the results of the literature review and add to the last paragraph of the introduction what your study contributes to existing literature when answering the main research questions.
Author response: As advised, we have indicated in the last paragraph of the Introduction what is lacking in the literature and have added our hypothesis, which if satisfied would allow us to identify and introduce influencing factors for nurse acceptance of telenursinginto the literature.As usual,the last paragraph of the Introduction must state the study objective, which we have also included. Please see the revised Introduction.
2.2 Review
2.2.1 Figure 1: There is some information missing in the box on behaviroal intention; please consider to revice the box on subjective norm (missing spaces)
Author response:Accordingly, Figure 1 has been extensively revised to include complete information. Please see the revised Figure 1.
2.2.2 Evidence on the TPB (Please note: this actually refers to subsection 2.4)
I am missing a bit the published literature on the evidence of TBP and the technology acceptance model for nurses attitudes towards digital technologies and, more specifically, telecare.
Please consider at least the following papers:
King WR, He J. A meta-analysis of the technology acceptance
model. Information & Management 2006; 43(6): 740-755..
Schlicht, L., Wendsche. J., Lehrke, L., Melzer, M., &Rösler, U. (2020). Nurses’ attitudes towards occupational transformation processes brought about by digital care technologies: Results from two cross-sectional studies. Current Directions in Biomedical Engineering, 6(3), 20203160. https://doi.org/10.1515/cdbme-2020-3160
The paper of Schlicht et al. will give you also some information on telecare acceptance and potential drivers in two German nurse samples (check alos the reference list for other studies on that topic).
Author response:
We have included additional references into the subsection 2.4Decomposed Theory of Planned Behavior in the Literature Review section 2.0 where TPB is discussed, including the suggested Schlicht et al. 2020 and several others. King et al. was used as the support for TAM. Please see the revised subsection 2.4.
2.3 Research questions and hypotheses
Under 2.5 please add at least research questions or better direceted hypotheses that fit with your analyses. You have the model in Figure 1 which is helpful here.
Otherwise, your analyses come a bit arbitrary and the reader those not really know what you are doing. Thus, such a section will give your paper more structure. Please note that this needs to give response to the RQs or Hypotheses in the results section.
Author response:As advised, we have added corresponding hypotheses to section 2.5 Research framework and research purpose. Please see the revised section 2.5 accordingly.
2.4 The research model in Figure 1
Is it possible to integrate the other relavent variables such as education and telecare experience in your research model? That would be a nice integration but also helps the reader to get a fast overview on the study variables.
Author response: Thank you for the suggestion. Moderators such as Knowledge and Barriers were considered in other models such as the Attitude/Social Norm/Self –efficacy model (ASE Model). We agree that moderator factors could be integrated to our model in the future. However, in the current model, “knowledge” was considered as part of “facilitating conditions.” items in the questionnaire (Supplemental Table S1) include:
"My nursing work institute/ unit has held telenursing related on-the-job education courses", and
"the work organization provides education and training on telenursing related knowledge and information systems"
Most importantly, our results showed that a stronger behavioral intention for practicing telenursing correlated with higher scores on these questions.
- Methods
3.1 Scales
Where do your scales and questions come from? Are these established scales or self-developed? This has to be reported (please add a reference for each scale, the number of items, a sample item) for each constructz and variable. Otherwise replication is not possible. If it is all self-developed, please add all scales and items in an appendix. What was the response format of the Likert scales?
Author response:The questionnaire was self-developed and included structured items based on the established model (attitude, subjective norms, perceived behavior control, behavioral intentions), and participants demographics. The construct measures were measured using a 5-point Likert scale. All items of the questionnaire are available as supplementary material. We have added this information to section 3.1 Research Tool. Please see the revised section 3.1 accordingly.
3.2 Construct Validity
I would like to encourage you to add further information on construct validity. This could be the results of an exploratory or better confirmatory factor analysis showing that all TPB variables are independent. This is also necessary since the intercorrelations in your study are extremly high.
Author response:This study adoptedexpert validity as the content validity index. The total content validity index (Total CVI) of the questionnaire was 0.93. Construct validity index was 0.91. This information was already included in the 3.1 “Research tools” subsection of the Methods section of our report. Please see the 3.1 “Research tools” subsection of Methods accordingly.
- Results
4.1 p. 9. first paragraph: I do not really understand what you are writing here. Do you mean that experienced nurses report higher usefulness, and so on? Please revise this section.
Author response:We have revised this paragraph to make it clearer for readers. Please see the revised results paragraph 4.1., as shown below:
Behavior (of practicing telenursing) for nurses with telenursing knowledge were positively influenced by perceived usefulness (t=2.519; p<0.05), subjective norms (t=2.082; p<0.05), superior influence (t=2.211; p<0.05), perceived behavior control (t=2.892; p<0.01), self-efficacy (t=3.135; p<0.01), and behavioral intention (t=2.762; p< 0.01).
4.2 Table 3: This is hard to read because you report so many decimals. From my persepective 2 decimals are enough.
Author response: As advised, Tables 2 and 3 have both been revised to show 2 decimal points only. Please see the revised Tables.
4.3 Variance inflation
Since your varaibles are all highly correlated you should report variance inflation factors (VIF) or, the better apporach, conduct a relative importance or relative weights analysis.
You find an online tool for such analyses here:
Tonidandel, S., LeBreton, J.M. RWA Web: A Free, Comprehensive, Web-Based, and User-Friendly Tool for Relative Weight Analyses. J Bus Psychol 30, 207–216 (2015). https://doi.org/10.1007/s10869-014-9351-z
The regression equation can be deleted (above table 5). Since you have some many tables I would also like to encoruageyoutor report the results of Table 5 in the text: "results of a multiple regression analysis with behavior intentions regressed on ... showed that ...". Why are the other predictors not integrated in that analysis such as usefulness, attioudes and so on. Form my perspecte the best way to test the model with ablockwise regression is:
Dependent Variable: Behavior Intentions
- Controls
- Attitudes, subjective norms and PBC (it is also to include the specific variables for the three factors blockwise)
Author response:
The regression above Table 5 was deleted as suggested. Also, the text results were amended as “The results of multiple regression analysis with behavioral intentions regressed on facilitating conditions and superior influence showed that “Behavioral intention = 7.543 + 0.394 x facilitating conditions + 0.232 x Superior influence”.」
In addition, the following response on VIF and relative weight analysis and a supplementary table has been added to subsection 3.8. Predictors of behavioral intention.Please see the revised section.
4.4 The paper of Schlicht et al found also an interaction effect of knwoledge and usability on attitudes for some digital technologies (but not for telecare). Is it possible to check this in a sublementaryanalysis.
Author response: In our study, “knowledge” and “usability” are considered as part of facilitating conditions, and items in the questionnaire include:
"the nursing staff work organization/institute has offered telenursing related on-the-job education courses"
"the work organization provides education and training on telenursing related knowledge and information systems" and
" fast response by the information department to users’ feedback or comments on the use of the information system’.
Our results showed that a stronger behavioral intention for practicing telenursing was associated with higher scores on these questions. These results were strong and we see no need for supplementary analysis.
- Discussion
5.1 The are some grammatical errors here. Please conduct proof reading.
Author response: The entire paper has been edited by a professional medical editor whose native language is English. We think you will find it more readable.
5.2 "The results of this study are similar to the results of a survey on obesity caregivers' exercise intentions." I would recommend to report on meta-analytical results on the TBP or on results regrading nurses attitudes towards digital technologies. Attitudes towards excercisebevaior are something different. This literature does not fit with your study (results).
Author response: We have deleted the reference as suggested. Other references are used to support our results. Please see the revised Discussion section.
5.3 What are the limitations of your study? Please add a section on this.
Author response: A limitation paragraph has been added at the end of the Discussion section. Please see the subsection 4.3 Limitations.
- Other Comments
- Please check the full manuscript for missing spaces between words (e.g. p.2., l 15: "TheCOVID-19 pandemichad" should be "The COVID-19 pandemic had.."; there are many of theses typos in the manuscript that need a throughly revision (check also the references; e.g. RogersM.Everett. Diffusion of Innovations: Third Edition. 1983.)
Author response:Thank you for picking up this formatting issue. We have made sure that such missing spaces are corrected in this revised manuscript.
Reviewer #2 Comments
In your article “The Intention and Influence Factors of Nurse’s Devotion of Telenursing”, you aim to understand the readiness of nurses to adopt telecare and identify factors that will support this process by questioning nurses and relate answers to the “Decomposed theory of plan behavior” and thus to theory on technology acceptance. Telecare is an important part of care, of which still much is to be learned, even though governments and health care organizations in different parts of the world, firmly promote telecare. In your article you bring forward the importance of nurses in telecare. To add on knowledge on nursing telecare, you conduct a quantitative research.
As I am not an expert on quantitative research, I restrain myself of comments on the methods of your study. I did comment on the rest of your paper though, and I am sorry to say that I have serious objections. I will list the different problems, with examples and advices:
- I find your article poorly structured. Throughout the whole article, new or repeated information on the field of nursing telecare and the importance of researching it are brought forward. Therefore, the pleas are hard to follow. To name an example: on page four, line 49, you give information on healthcare costs, in relation to self-management (which I feel is a misused argument there anyway) in a section on ‘The performance on nursing staff in telecare’. I find it very hard to follow how this information fits that plea, and it happens throughout the entire article. Re-structure your article, carefully building your argument on the importance of your article by re-organizing and replacing the different parts
Author response: We have taken your comments into consideration and have completely re-edited the text, deleting much of misplaced or redundant material and improving succinctness.We have simplified some of the structural aspects but have maintained Methods, Results and Discussion as discrete sections. We hope the extensive editing will improve readability as well as content.
- I strongly advise you to make sure that the text is edited by a professional English editor.
Author response: The entire paper has been edited by a professional medical editor whose native language is English. We think you will find it more readable.
- Be very precise on the concepts you use, be consistent and support them with appropriate literature. For example, telecare, telehealth, and virtual communication – they are not the same and need explanation. You also introduce some concepts that may determine your research to a great extent, like ‘implementation’, without supporting them with literature.
Author response: Telehealth is a general subject and telenursing is a specific subset of telehealth. We have rewritten the Introduction to explain this more clearly and have supported it with references. We have again reviewed the entire manuscript for inconsistency and have tried to use telenursing throughout when we are speaking specifically about nursing adoption of telehealth. However, sometimes when we cited a specific reference, we used the authors’ terms. We hope you fill find terminology more consistent.
- The inconsistency of concepts, also troubles the reader in understanding the results. The advice is once again to be precise in the introduction of concepts in the beginning and use them till the end of the article.
Author response: As explained above, telehealth is a general subject and telenursing is a specific subset of telehealth. We have rewritten the Introduction to explain this more clearly and have supported it with references. When introducing or discussing concepts in the Introduction, Methods and Discussion, we have provided more references and have rewritten those sections to clarify these concepts for the reader.. Please see the revised text.
- In your conclusion, you state “As COVID-19 remains a pandemic, telecare can reduce costs, improve quality, and reduce cross-infection”. I would advice you to broaden your literature review and relate to scholars from nursing and STS, who have more layered and nuanced views on the predictability of telecare. This can add to the importance of your advices and conclusions.
Author response: We appreciate your comment, we have added an appropriate COVID reference as suggested as well as other up-to-date references on nurse acceptance of telenursing. However, our conclusion cannot discuss COVID-19 in depth,especially because we did not investigate telehealth aspects of COVID-19 care. Our conclusion focuses on what we learned about factors that influence nurses’ behavioral intentions to use telenursing.

Reviewer 2 Report
Dear authors,
In your article “The Intention and Influence Factors of Nurse’s Devotion of Telenursing”, you aim to understand the readiness of nurses to adopt telecare and identify factors that will support this process by questioning nurses and relate answers to the “Decomposed theory of plan behavior” and thus to theory on technology acceptance. Telecare is an important part of care, of which still much is to be learned, even though governments and health care organizations in different parts of the world, firmly promote telecare. In your article you bring forward the importance of nurses in telecare. To add on knowledge on nursing telecare, you conduct a quantitative research.
As I am not an expert on quantitative research, I restrain myself of comments on the methods of your study. I did comment on the rest of your paper though, and I am sorry to say that I have serious objections. I will list the different problems, with examples and advices:
- I find your article poorly structured. Throughout the whole article, new or repeated information on the field of nursing telecare and the importance of researching it are brought forward. Therefore, the pleas are hard to follow. To name an example: on page four, line 49, you give information on healthcare costs, in relation to self-management (which I feel is a misused argument there anyway) in a section on ‘The performance on nursing staff in telecare’. I find it very hard to follow how this information fits that plea, and it happens throughout the entire article. Re-structure your article, carefully building your argument on the importance of your article by re-organizing and replacing the different parts
- I strongly advise you to make sure that the text is edited by a professional English editor.
- Be very precise on the concepts you use, be consistent and support them with appropriate literature. For example, telecare, telehealth, and virtual communication – they are not the same and need explanation. You also introduce some concepts that may determine your research to a great extent, like ‘implementation’, without supporting them with literature.
- The inconsistency of concepts, also troubles the reader in understanding the results. The advice is once again to be precise in the introduction of concepts in the beginning and use them till the end of the article.
- In your conclusion, you state “As COVID-19 remains a pandemic, telecare can reduce costs, improve quality, and reduce cross-infection”. I would advice you to broaden your literature review and relate to scholars from nursing and STS, who have more layered and nuanced views on the predictability of telecare. This can add to the importance of your advices and conclusions.
Author Response

(The authors gave the same response as above.)
